# Similarity Changes Analysis for Heart Rate Fluctuation Regularity as a New Screening Method for Congestive Heart Failure

**DOI:** 10.3390/e23121669

**Published:** 2021-12-11

**Authors:** Zeming Liu, Tian Chen, Keming Wei, Guanzheng Liu, Bin Liu

**Affiliations:** 1School of Biomedical Engineering, Shenzhen Campus of Sun Yat-sen University, Shenzhen 518107, China; zemingliuics@163.com (Z.L.); chent295@mail2.sysu.edu.cn (T.C.); weikm3@mail2.sysu.edu.cn (K.W.); 2School of Science, Hua Zhong Agricultural University, Wuhan 430070, China

**Keywords:** congestive heart failure (CHF), autonomic nervous system (ANS), heart rate variability (HRV), fuzzy approximate entropy of similarity-based information (fApEn_IBS)

## Abstract

Congestive heart failure (CHF) is a chronic cardiovascular condition associated with dysfunction of the autonomic nervous system (ANS). Heart rate variability (HRV) has been widely used to assess ANS. This paper proposes a new HRV analysis method, which uses information-based similarity (IBS) transformation and fuzzy approximate entropy (fApEn) algorithm to obtain the fApEn_IBS index, which is used to observe the complexity of autonomic fluctuations in CHF within 24 h. We used 98 ECG records (54 health records and 44 CHF records) from the PhysioNet database. The fApEn_IBS index was statistically significant between the control and CHF groups (*p* < 0.001). Compared with the classical indices low-to-high frequency power ratio (LF/HF) and IBS, the fApEn_IBS index further utilizes the changes in the rhythm of heart rate (HR) fluctuations between RR intervals to fully extract relevant information between adjacent time intervals and significantly improves the performance of CHF screening. The CHF classification accuracy of fApEn_IBS was 84.69%, higher than LF/HF (77.55%) and IBS (83.67%). Moreover, the combination of IBS, fApEn_IBS, and LF/HF reached the highest CHF screening accuracy (98.98%) with the random forest (RF) classifier, indicating that the IBS and LF/HF had good complementarity. Therefore, fApEn_IBS effusively reflects the complexity of autonomic nerves in CHF and is a valuable CHF assessment tool.

## 1. Introduction

Congestive heart failure (CHF) is a clinical condition with inadequate ventricle filling or manifestation of inadequate myocardial contraction (myocardial failure), caused by changes in cardiac structure and function [1]. It has been identified as a major public health concern and extensively studied within the past two decades. There are about 3 million patients in the United States, nearly 1.5% of the adult population suffering from CHF [2]. The amount of CHF patients is rapidly growing because of the population aging and the increase of survival after myocardial infarction [3]. In addition, CHF may also cause some underlying heart disease [4], or bring structural and functional derangements to the liver with congestion [5]. Thus, the diagnosis of CHF patients is critical.

A previous study proved that cardiac arrhythmias in CHF is associated with the instability of autonomic nervous system (ANS) [6]. To this end, scholars usually extract heart rate variability (HRV) indicators from ECG signals, which can reflect the function of ANS, such as the vagal and sympathetic function, to estimate the activity of ANS in CHF [7,8]. Traditionally, HRV analysis is usually based on frequency and time domain. Takase et al. [9] reported that the HRV index on time-domain could successfully distinguish patients and normal individuals. Rovere et al. [10] found that power in low-frequency LF (0.04–0.15 Hz) can effectively predict the probability of sudden death in patients with CHF. However, these methods are effective, HR fluctuations are considered complex behaviors derived from nonlinear and non-stationary processes [11,12,13]. So, the traditional linear methods may not suit CHF studies because they would be disturbed by fluctuations and cause experimental errors.

Therefore, many nonlinear HRV analysis methods have been extensively studied in recent years, including the fractal dimension [14], Poincare diagram [15], complexity analysis [16] and other methods, all of which have achieved certain results. However, Li et al. [17] pointed out that few scholars have studied the dynamic changes of short-term HR fluctuations, and most of them only evaluated the overall level of HRV. Nevertheless, information-based similarity (IBS) was proved to be a valuable way of nonlinear methods to study HRV [18,19], many scientists demonstrated that some underlying dynamics existed in the complex fluctuations in the output signals of the physiological system [12,20]. IBS method considers nonlinear dynamic characteristics of a physiological signal. It studies the acceleration/deceleration mode of heart rate fluctuations based on the sequencing frequency analysis. By transforming HR fluctuations into symbols, it can effectively correct the noise made by the antagonism effect of sympathetic nerve system (SNS) and parasympathetic nervous system (PNS) effectively [21]. In this way, the IBS method can be applied to get the instantaneous HR fluctuation differences between CHF patients and normal. Cui et al. [21] found that IBS index could correctly classify Atrial fibrillation (AF) and normal heart rate. Yang et al. [22] used IBS to study β-AR polymorphism, which reflected the underlying pattern of ANS and the difference between ANS changes and dynamic heart rate changes.

Entropy is also a feasible method for CHF study for the complexity of heart rate fluctuations [23,24]. Many studies proved the effectiveness of entropy in CHF screening [25], which supported that the HRV analysis was helpful in CHF diagnosis. Moreover, introducing the concept of “fuzzy set”, fuzzy approximate entropy (fApEn) is an ideal tool to measure the complexity of time series [26]. Li et al. [27] indirectly processed ECG signals to expose ANS imbalance in OSA patients by introducing fuzzy approximate entropy to calculate the complexity of slope series, rather than directly calculating the entropy value of the ECG data. In this way, the fApEn could indirectly estimate the complexity of HR fluctuations during a short time period. Sokunbi et al. [28] found that fApEn not only depends heavily on data length, but also has a better performance in monotonicity, relative consistency, and robustness to noise than other entropy methods when processing different complexities signals.

Therefore, the combination of IBS and fApEn cannot only reveal the law of HR fluctuations but also better study sympathetic disorders by measuring the complexity of different wave patterns, and also reveal the mechanism of CHF patients. First, the IBS method was used to convert the HRV signal into an IBS value. Secondly, fApEn_IBS was extracted to study the complexity of short-term heart rate fluctuations. In addition, LF/HF was also calculated as the contrast, the significance analysis of the three indicators was conducted. Finally, the Fisher discriminant function was used for single index discrimination. Since machine learning methods have been proven to be able to successfully detect and classify heart failures [29,30,31], we decided to use machine learning classifiers for multi-feature discrimination to improve screening accuracy. The samples were divided into normal and CHF groups, and the accuracy, specificity and sensitivity were calculated.

## 2. System and Method

In this study, HRV characteristics of normal and CHF subjects were analyzed. The framework of the system is shown in Figure 1. Firstly, the 24-h RR-interval signals were extracted after collecting ECG signals. Then, the corrected 1-min RR-interval signals (RRI) were obtained through preprocessing. Furthermore, the traditional HRV research methods and the IBS method were used to calculate the indices. Finally, LF/HF ratio and IBS index were used for statistical analysis and severe CHF detection.

### 2.1. Data Collection

In this study, the 24-h RR interval signals of 54 healthy subjects (31 males and 23 females, aged 61.38 ± 11.63 years) were collected from the Normal Sinus Rhythms RR Interval database, and 44 CHF subjects (19 males and 6 females, 19 subjects’ gender were unknown, aged 55.51 ± 11.44) were acquired from the Beth Israel Deaconess Medical Center (BIDMC) Congestive Heart Failure database (15 subjects) and the Congestive Heart Failure RR Interval database (29 subjects). All these databases and ECG signals are open source from the PhysioNet open database [32,33,34].

### 2.2. Preprocess

To reduce the affect produced by noise, the 24-h HRV signals were preprocessed. First, we remove the first and last RR intervals of each 24-h record, as well as the RR intervals, which are longer than 3 s, to remove singular value interference [35]. Then, we did not choose to delete lower outliers because the RR intervals of healthy groups are usually higher than CHF groups [9]. Many researchers considered 5-min series as a standard interval of spectral HRV studies [36]. However, based on the situation the 1-min segments would have a good real-time performance for efficiently capturing changes in heart rate. In this way, RRI may also be suitable for CHF detection [37] and we separated the 24-h HRV signals into non-overlapping 1-min RRIs.

### 2.3. Indices Calculation

#### 2.3.1. Frequency Domain Analysis

In frequency domain, the low-frequency power (0.04–0.15 Hz, LF) is closely related to the SNS activity, and the high-frequency power (0.15–0.4 Hz, HF) is usually associated with the activity of the PNS. Then, the LF to HF ratio (LF/HF) can reflect the tension balance between SNS and PNS [38]. In this study, a fast Fourier transform would be used to compute the power spectral density for each RRI. The LF/HF ratio was calculated by the LF and HF components index, which is widely accepted to reflect the ANS balance [39]. The formula is shown as:(1)$$\mathrm{LH}\;\mathrm{Ratio}=\frac{\mathrm{LF}\;\mathrm{power}}{\mathrm{LH}\;\mathrm{power}}$$

#### 2.3.2. Similarity Change Analysis

In the previous studies, information-based similarity (IBS) of RR-interval signals can successfully analyze the similarity of the regularity in short-term HR fluctuation [21,40]. Then, fuzzy approximate entropy (fApEn) can reflect the complexity of short-term HR fluctuations’ similarity. Based on similarity analysis and entropy measurement, the fApEn_IBS was proposed as a novel HRV analysis method to determine the dynamic rhythm changes of the ANS by observing the complexity changes of fluctuation patterns. The scheme of the fApEn_IBS method is shown in Figure 2, and the details of the fApEn_IBS are as follows.

Step 1 (RRI reconstruction): the raw RRI is defined as follows:(2)$$\mathrm{RRI}=\left(r_{1},r_{2},r_{3},\ldots ,r_{k}\right)\;k=1,2,\ldots ,n$$

We assume that the coarse-grained is s, that is, the data of every s RR interval will be classified into a set for non-overlapping mean coarse-grained processing. This means that you will get a new variable y_j_, whose value is the average of the data in the corresponding set:(3)$$y_{j}=\frac{\sum_{k=\left(j-1\right)*s+1}^{j*s}r_{k}}{s}.\;j=1,2,\ldots ,\;n^{\prime }.$$

Through non-overlapping mean coarse-grained processing, the original RRI sequence is reconstructed into a new sequence Y, which is formed by y_j_:(4)$$Y=\left(y_{1},y_{2},\ldots ,y_{\;n^{\prime }}\right)\;n^{\prime }=\left[\frac{n}{s}\right].$$

Figure 3 illustrates the details of RRI reconstruction. As shown in Figure 3, the first s RR interval data r_1_ to r_s_ are classified into the first set, and y_1_ is obtained after averaging these s data. By analogy, the new sequence Y composed of y_1_ to y_n_ replaces the original RRI sequence.

Step 2 (Construction of m-bit words sequence): The increase and decrease between the y_i_ and y_i+1_ were defined as 1 and 0, respectively. Based on this standard, the Y sequence with n′ length would be converted into a 0–1 binary sequence of length n′-1. Then, a binary sequence of length m is represented as a word, called a m-bit word. Each different m-bit word could reflect a specific pattern of HR fluctuations [40]. In this way, by applying the sliding-window method to Y, it became a series of m-bit words.

Step 3 (Distance calculation): By quantifying the number of each word in the sequence, the frequency of occurrence of different words can be obtained. Moreover, finally, every word’s position in the sequence would depend on its frequency after sorting. To calculate the IBS value between sequences of adjacent words, the formulas are as follows [21]:(5)$$w\left(x_{i}\right)=\frac{\;\left[-p_{1}\left(x_{i}\right){\log\;p}_{1}\left(x_{i}\right)-p_{2}\left(x_{i}\right){\mathrm{logp}}_{2}\left(x_{i}\right)\right]}{\sum_{i=1}^{L}\left[p_{1}\left(x_{i}\right){\mathrm{logp}}_{1}\left(x_{i}\right)-p_{2}\left(x_{i}\right){\mathrm{logp}}_{2}\left(x_{i}\right)\right]}$$
where w is the weight of a decimal m-bit word x_i_, L represents the total number of different m-bit words, and p denotes the possibility of each word.
(6)$$D\left(R_{1},R_{2}\right)=\frac{\sum_{i=1}^{L}\left|K_{1}\left(x_{i}\right)-K_{2}\left(x_{i}\right)\right|w\left(x_{i}\right)}{L}$$

D(R1, R2) denotes the distance between adjacent words sequences, as well as the IBS value between adjacent raw RRIs. K denotes the rank of word.

Step 4 (Average calculation): After calculating the average of all IBS values between adjacent RRIs in one recording, the IBS index would be obtained.

Step 5 (Space reconstruction & Space vectors distance calculation): From the time series containing n IBS values in one recording u(t) = u(1), u(2), ..., u(N), the following m-dimensional vector is defined as:$${\mathrm{XX}}_{i}^{m}=\left\{u\left(i\right),\ldots ,u\left(i+m-1\right)\right\}-u_{0}\left(i\right)$$
(7)i=1,2,…,n−m+1
where $u_{0}\left(i\right)$ is calculated as:(8)$$u_{0}\left(i\right)=\frac{\sum_{k=0}^{i+m-1}u\left(k\right)}{m}$$

Distance $d_{\mathrm{ij}}^{m}$ between vectors ${\mathrm{XX}}_{i}^{m}$ and ${\mathrm{XX}}_{j}^{m}$ is based on maximum absolute difference calculation. For a given similarity tolerance of function r, the similarity degree $D_{\mathrm{ij}}^{m}$ is determined by a fuzzy membership function FZ($d_{\mathrm{ij}}^{m}$, r), which employs the Gaussian function.
(9)$$D_{\mathrm{ij}}^{m}=\mathrm{FZ}\left(d_{\mathrm{ij}}^{m},r\right)$$

Step 6 (Approximation calculation): define the possibility of vectors ${\mathrm{XX}}_{i}^{m}$ and ${\mathrm{XX}}_{j}^{m}$ will match as function $\varphi_{i}^{m}$,
(10)$$\varphi_{i}^{m}=\frac{\sum_{j=1}^{N-m}D_{\mathrm{ij}}^{m}}{N-m+1}$$

Moreover, define the function $\Phi^{m}\left(r\right)$ as follows:(11)$$\Phi^{m}\left(r\right)=\frac{\sum_{i=1}^{N-m+1}\ln\left(\varphi_{i}^{m}\right)}{N-m+1}$$

Because the input IBS parameter is a finite time series [41],
(12)$$\begin{array}{l} \mathrm{fApEn}\left(m,r,N\right)=\Phi^{m}\left(r\right)-\Phi^{m+1}\left(r\right)\\ =\ln\frac{{\left(N-m+1\right)}^{-1}\sum_{i-1}^{N-m+1}\varphi_{i}^{m+1}}{{\left(N-m\right)}^{-1}\sum_{i-1}^{N-m}\varphi_{i}^{m}} \end{array}$$

Based on these, fApEn_IBS value could be calculated.

### 2.4. Indice Validation

To evaluate the HRV indices, first we calculate LF/HF ratio and IBS index as well as fApEn_IBS index based on MATLAB (R2017b, MathWorks, Natick, MA, USA). Then, a t-test was used to analyze the significant difference between the control and CHF groups based on SPSS version 22.0.0.0 (SPSS Inc., Chicago, IL, USA), the results were expressed as the mean ± SD. Moreover, *p* < 0.05 was considered statistically significant. Finally, to verify the performance of these HRV indicators, Fisher linear discriminant of SPSS was used for CHF screening. In addition, based on the Python 3.6.5 environment, we use support vector machine (SVM) [42], k-nearest neighbor (KNN) [43], and Random forests (RF) [44] to perform five cross-validation for multiple features. The results were expressed as accuracy, sensitivity, and specificity. The receiver operating characteristic (ROC) curve and area under the ROC curve (AUC) of the algorithms were calculated using MATLAB (R2017b, MathWorks, Natick, MA, USA).

## 3. Results

### 3.1. HRV Analysis of the Difference between the Control and CHF Group

The analysis results of the frequency domain, IBS and fApEn_IBS indices are shown in Table 1 as mean ± SD values. There were significant differences between the healthy and CHF groups (*p* < 0.001) in the three indices, so LF/HF and IBS, fApEn_IBS indexes are all considered as useful indices to distinguish the control and CHF groups.

Figure 4 shows the representative adjacent raw RRIs between the normal and CHF group. Moreover, it also reveals that the regularity of HR fluctuations between adjacent RRIs in the two groups was different, the regularity of HR fluctuations in CHF group is more similar than normal people. Moreover, as shown in Figure 5, two records were selected from the normal and CHF groups as typical examples to reflect the distribution of IBS index in 24 h. The complexity of color distribution indicated that the degree of disturbance of HR fluctuation was irregular between the two groups.

### 3.2. CHF Sreening

Using the Fisher linear discriminant function, each sample was classified as normal and CHF patient to access the performance of each index. The accuracy, sensitivity and specificity are defined as the percentage of correctly classified normal or CHF patients, the percentage of correctly classified CHF patients, and the percentage of correctly classified normal samples, respectively. Table 2 shows that fApEn_IBS is the best screening indicator with the highest accuracy (84.69%) and specificity (98.15%) for 98 samples. Therefore, fApEn_IBS improves detection accuracy and is a more useful index than LF/HF and IBS. As shown in Figure 6, two groups could be separated from three-dimensional observations, proving the rationality of screening by three indicators.

When the samples have multiple features and a large amount of data, machine learning classifiers will have significant advantages over other methods [45]. We selected the following three machine learning methods for multi-index screening:(1)K Nearest Neighbor (KNN): KNN is a type of instance-based learning. New cases are classified based on a similarity measure in the vector space model. Here the neighbor number is defined as 5 [46].(2)Random forest classifier (RF): RF is an ensemble learning method that consists of a multitude of decision trees. The result is determined by the output model of a single tree [47].(3)Support Vector Machine (SVM): SVM is a supervised clustering method that maps data points to high-dimensional space through kernel functions for classification. In this paper, we select polynomial kernel function to classify data points [48].

Combining the above five-fold cross-validation analysis, the screening effect is better. With multiple features, including IBS, LF/HF, and fApEn_IBS index, the random forest classifier achieved the best classifier result, as shown in Table 3. Compared with other classifiers, RF achieved a better classification performance under the same input conditions, with the highest accuracy and more balanced sensitivity and specificity (98.98% accuracy, 98.15% sensitivity and 100% specificity). The confusion matrix also points out that only one CHF patient was wrongly identified, which proved that the RF classifier is an effective method for CHF detection. Furthermore, Figure 7 also supports this conclusion through the ROC curve that is the closest to the top with the largest area (AUC = 0.9996).

### 3.3. Parameter Selection

When calculating the fApEn_IBS index, the influences of different bits of constructed words and the coarse-grained scale s on the CHF screening performance are shown in Figure 8. When s and m are 2 at the same time, the screening performance is optimal. Therefore, s = 2 and m = 2 are selected as the average coarse-granulation scale and the word length of the construction. When m = 2, it can better capture the relationship between adjacent letters, and different numbers are more likely to appear. When s = 2, the information of the original RR interval sequence can be preserved more finely and noise would be eliminated. In conclusion, when s = 2 and m = 2, CHF detection performance is the best and screening accuracy is improved. So, m = 2 and s = 2 are selected as the number of bits of the words and coarse-grained scale to calculate the fApEn_IBS index for HRV analysis.

## 4. Discussion

### 4.1. Comparison and Summary

Previous studies have shown that LF/HF is a useful frequency domain indicator for HRV analysis in CHF patients [49,50], and our results also confirmed that there is a statistical difference in LF/HF between healthy people and CHF patients (*p* < 0.001). However, as the study progressed, the screening accuracy of the LF/HF indicator was not high enough (less than 80%), the single index screening accuracy of IBS and fApEn_IBS was higher.

Furthermore, fApEn_IBS has the most significant performance in distinguishing normal people and CHF patients. Compared with the IBS index, fApEn_IBS can not only analyze the similarity of HR fluctuation, but also analyze the change of fluctuation regularity. Table 2 shows that the accuracy of fApEn_IBS in CHF screening is 84.69%, while the screening accuracy of IBS, LF/HF is 83.67% and only 77.55%, respectively. In addition, the specificity and sensitivity of fApEn_IBS were 98.15% and 68.18%, respectively. The high specificity indicates that fApEn_IBS has strong screening ability for patients with congestive heart failure, and its low sensitivity may be due to the smaller number of positive patients than normal people, so small fluctuations in the number have a greater impact on the calculation of the index. In summary, the fApEn_IBS index is a valid index for CHF screening.

### 4.2. Comparison with Previous Studies

To expand the sample size, we collected data from three databases. Focusing on study methods and screening results, we compared them with related studies that analyzed ECG data [25,51,52,53,54], as shown in Table 4.

Some of the previous studies only focused on the screening by nonlinear methods [25,51] and did not combine machine learning with improving the screening accuracy. Other studies had smaller sample sizes or inadequate screening accuracy [52,53]. Compared with the results of other studies, the classification accuracy of our method is higher than most with the accuracy (99%), sensitivity (97.8%) and specificity (100%) all above 95%. These results indicate that fApEn_IBS has a certain clinical reference value in evaluating ANS complexity for CHF patients.

### 4.3. Method Propsed and Parameter Selection

A previous study has reported that 5-min series is the standard time interval for HRV studies [36], but considering that RRI could be keenly aware of heart rate changes, the 1-min segment has good real-time performance. Although the traditional classical frequency-domain method [10] is feasible, the regularity and complexity of HR fluctuations are considered more comprehensively. Therefore, the fApEn_IBS index was constructed by a 1-min RR segment for CHF diagnosis in this paper.

It is found that the interval time series of heartbeats are correlated [55], and it can be changed with some diseases [56]. As shown in Figure 6, CHF patient has more similar HR fluctuations and a higher correlation between adjacent RRIs than normal subject. Therefore, correlation analysis can be used to detect CHF. However, most studies did not quantify the degree of correlation. As a method to measure the similarity between symbolic sequences, IBS had been used effectively to quantify the correlation between interbeat interval time series [18,40].

As a nonlinear system, the function of the cardiopulmonary coupling system is affected by CHF, and the regulation of the system to HR through ANS is also nonlinear [57]. Considering that HRV also contains nonlinear components in signal generation and regulation, compared with the linear method, more information can be obtained by processing HRV signal with the nonlinear method [58]. IBS as a novel nonlinear method, before and after using the numbers reflect the change of HR fluctuations, thus weakening the influence of volatility. Moreover, after converting the 0–1 binary sequences into m-bit words sequences, IBS is only associated with the relative proportion of letters that appeared instead of the absolute number and time node of occurrences of letters. Therefore, the IBS method can analyze the similarity of nonlinear system signals by comparing the variation rule, and this is a reasonable strategy to detect CHF by evaluating the similarity of HR fluctuation regularity.

Complexity is a feature of normal cardiovascular regulation, and the nonlinear regulatory pattern of the cardiovascular system leads to the complexity and irregularity of RRI [59]. As a complexity analysis method, fuzzy approximate entropy is not strongly dependent on data length when processing different complex signals, and has better performance in terms of robustness to noise [28]. Thus, after studying the HR fluctuation rule by IBS sequences, the complexity of IBS sequence would be studied by combining fuzzy approximate entropy algorithm, and the fApEn_IBS index was constructed to observe the complexity difference HR fluctuation between normal people and CHF patients.

To improve index performance and detection accuracy, the parameters of different bits of constructed words and the coarse-grained scale and fApEn were also analyzed. Wu et al. [40] revealed that if the constructed words in sequences for RRI are too long, there would be too few m-bit words. Therefore, we calculated the IBS index when M was 2–6. If the degree of coarse granitization is too large, NAN value will be generated and noise interference cannot be effectively eliminated. In addition, considering the result of retaining original sample data when s value is 1, the value range of S is 1–7. Three parameters of fApEn must be determined: parameter N, which represents the number of sequence points; M is the embedding dimension of sequence reconstruction; r is the similarity tolerance of the exponential function. Sun, Zheng et al. [60,61] proved the feasibility of the method when the parameters N, M and r were set to 2, 2 and 0.25, respectively, and the results were consistent with other results.

### 4.4. Physiological Significance

CHF is a chronic cardiovascular syndrome associated with ANS dysfunction [6], HRV is a useful tool for evaluating ANS [7,8], and previous studies on the changes of HRV signal energy have attempted to screen CHF patients. In this paper, three indexes of LF/HF, IBS and fApEN_IBS were constructed to analyze the changes of HRV signal by evaluating the balance of ANS function, the similarity and complexity of heart rate fluctuation rhythm. Finally, according to *p*-value in the Table 1, it was concluded that there were significant differences in three aspects between patients and healthy people.

#### 4.4.1. Why Does ANS Balance Differ Significantly between the Two Groups?

The traditional HRV frequency domain method, LF/HF, reflects the balance of ANS; that is, the level of sympathetic and parasympathetic nervous system activity, demonstrating a significant difference between the normal group and CHF group in our study. Valenza et al. [62] indicated that vagal activity was significantly reduced in patients with CHF. Kishi et al. [63] found an imbalance between the sympathetic nerve and the vagus nerve during a CHF attack. Hasking et al. [64] proposed that to compensate for the pumping function of the heart, patients with CHF increase the level of norepinephrine through a vicious cycle, thereby increasing the sympathetic outflow of the heart and leading to more sympathetic nerve activity. These are the reasons for the statistically significant difference in LF/HF between the two groups, which show a significant difference in ANS balance.

#### 4.4.2. Why Does the Similarity of HR Fluctuations Change Significantly between the Two Groups?

Rahko et al. [65] found that systolic cardiac function was obstructed in patients with CHF, characterized by a significant abnormality in the total systolic interval. Moreover, increased sympathetic activity is accompanied by increased heart rate and energy expenditure, resulting in decreased myocardial oxygen delivery, gradual delay and deceleration of diastole [66,67], which is closely associated with malignant arrhythmias [68]. All these contribute to decreased ability of the autonomic nerve to regulate the change of heart rate and the slow fluctuation of heart rate, leading to the significant difference in the similarity of HR fluctuation rule between the two groups. Figure 6 also confirms that patients’ adjacent RR interval fluctuation is more similar.

Understanding from each point in time, IBS reflects the transient fluctuation of heart rate; understanding from a period of time, and IBS reflects the similarity of the HR fluctuation rule of the sample. CHF patients had lower IBS values, indicating that the adjacent RR intervals fluctuated less and the fluctuation pattern was more similar. With the decrease of oxygen intake, patients cannot change cardiac output by heart rate regulation. The heart cannot fully discharge the venous return of blood from the heart, leading to blood stasis in the venous system and blood perfusion in the arterial system [69]. This increased atrial pressure, which in turn leads to increased capillary pressure, causes cardiac circulation disorders such as congestion in the lungs, and thus patients with CHF are less able to adapt to their environment [70,71].

#### 4.4.3. Why Does the Complexity of HR Fluctuations Differ between the Two Groups?

Saul et al. [72] found that during heart failure, vagal activity decreased. However, sympathetic modulation relatively increased, and these findings reflect that the regulation of HR in CHF patients was significantly disorganized. Naturally, there is a difference in the complexity of HR fluctuations between patients and normal, but dysregulated HR regulation does not mean great complexity. Zhao et al. [73] pointed out that CHF subjects would lose complexity due to pathology. Yin et al. [74] found that the nonlinear complexity of the healthy group was higher than that of the CHF group because the complexity of the cardiovascular system of patients would be reduced, and the entropy value of the complexity of the ECG description would be correspondingly reduced. All of these conclusions are consistent with our experimental results.

The fApEn_IBS of CHF patients was smaller, indicating that the complexity of short-term HR fluctuations in the 24 h was reduced, reflecting that patients are accompanied by impairment of cardiac function and decline of ventricular systolic and diastolic functions [75], and that ANS injury is severe and imbalanced, leading to a relatively simple pattern of short-term heart rate fluctuations. Autonomic nerves do not effectively regulate HR to increase cardiac output and restore systolic/diastolic function [69].

Limitation:

There are some limitations to this study. Firstly, more databases should be used to expand the sample size to verify the effect of the method. Secondly, we only studied the validity of 1-min clips, and further studies on the length of time should be conducted in the future. Thirdly, gender and age inconsistencies between the CHF and control groups may also lead to differences in experimental results. Finally, the effect of drug use on CHF patients should be valued and quantified, excluding the interference of irrelevant variables, so this limitation should be considered in future studies.

## 5. Conclusions

In this study, the fApEn_IBS method was used to extract the information of different RR intervals. This method not only evaluated the similarity of the HR fluctuation regularity between adjacent RR intervals, but also reflected the complexity of the fluctuation rule, thus significantly improving the CHF screening performance. Our results showed that the similarity and complexity of HR fluctuation law between adjacent RRI in the CHF group were significantly reduced, and the accuracy of fApEn_IBS in CHF detection was 84.69%, and the accuracy of fApEn_IBS, IBS and LF/HF combined with random forest classifier was 98.98%. Therefore, the fApEn_IBS method can be used to analyze the changes of short-term HR fluctuation regularity and detect the patients with CHF.

## Figures and Tables

**Figure 1 entropy-23-01669-f001:**
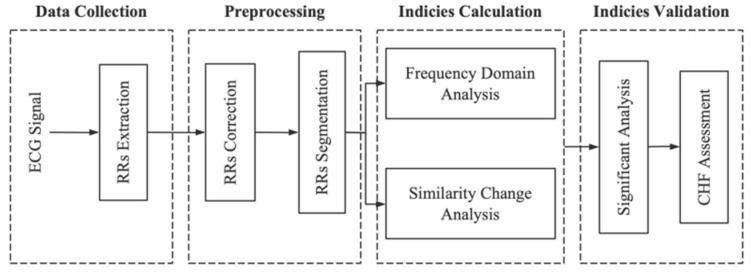
Framework of the proposed HRV analysis system.

**Figure 2 entropy-23-01669-f002:**
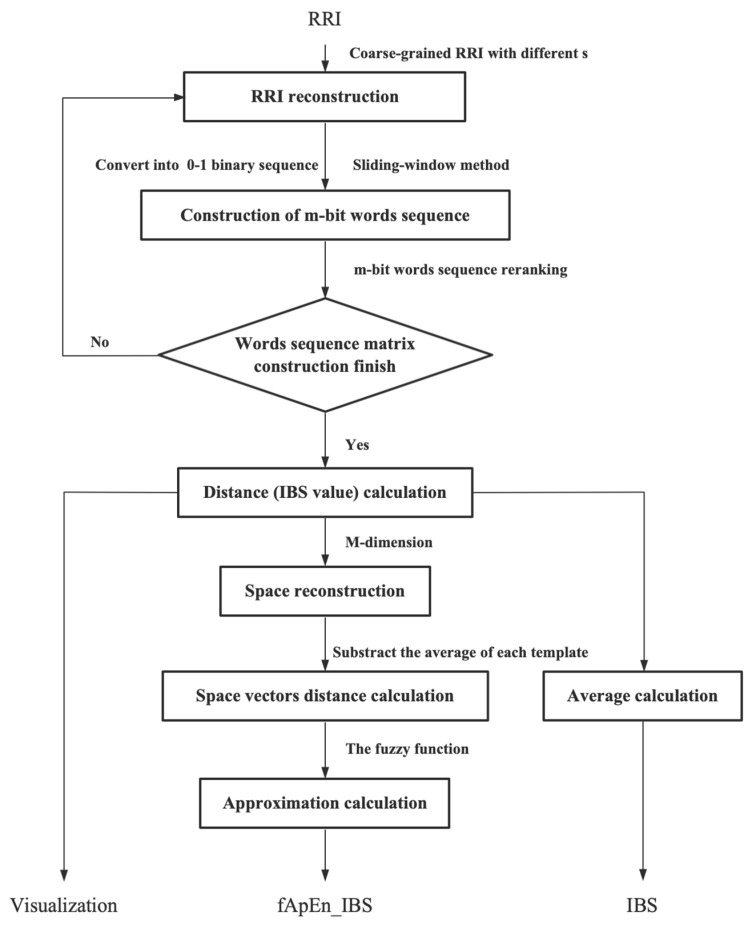
Scheme of similarity change analysis.

**Figure 3 entropy-23-01669-f003:**
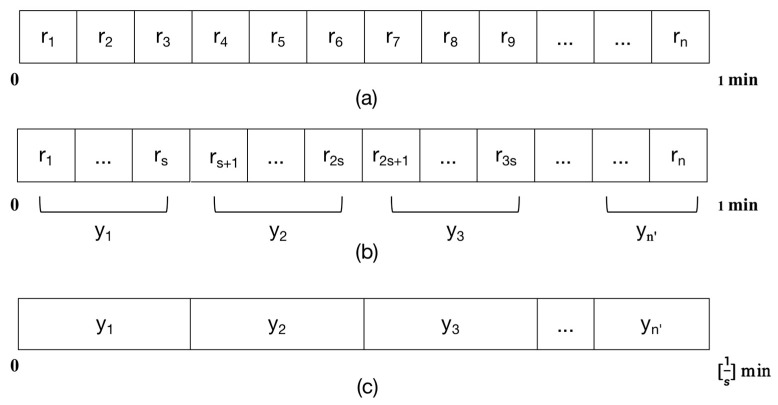
An illustration of RRI reconstruction. (**a**) A 1-min RRI = $\left(r_{1},r_{2},r_{3},\ldots ,r_{n}\right)$. (**b**) Multiple non-overlapping r_ks_ were divided from RRI. (**c**) The reconstructed sequence Y is composed of a time duration of 1/s minute.

**Figure 4 entropy-23-01669-f004:**
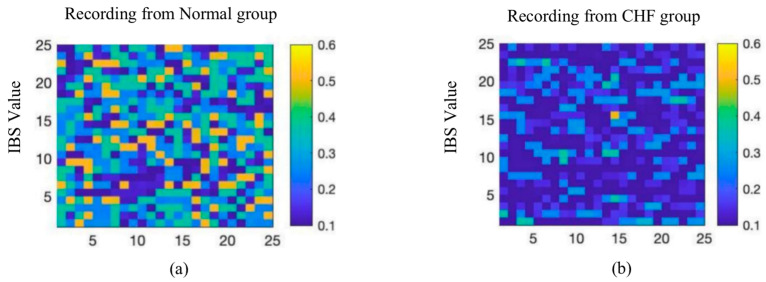
Adjacent raw RRIs of a recording from the Normal group (**a**), the CHF group (**b**). The blue line represents the RR_(i−1)_, the gray line represents the RR_(i)_, (2 ≤ i ≤ n, and n is the total amount of RRI in the selected recordings).

**Figure 5 entropy-23-01669-f005:**
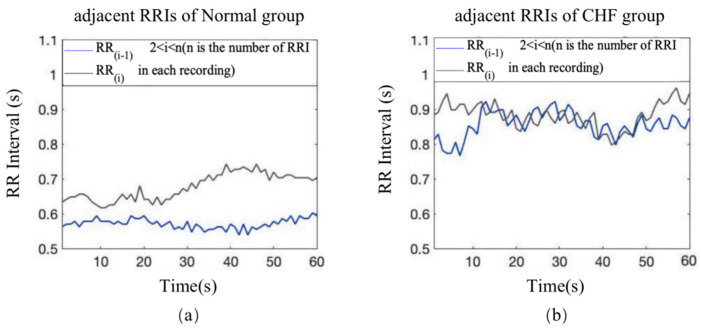
Distributions of 24-h period IBS index of a typical subject from the Normal group (**a**), CHF group (**b**).

**Figure 6 entropy-23-01669-f006:**
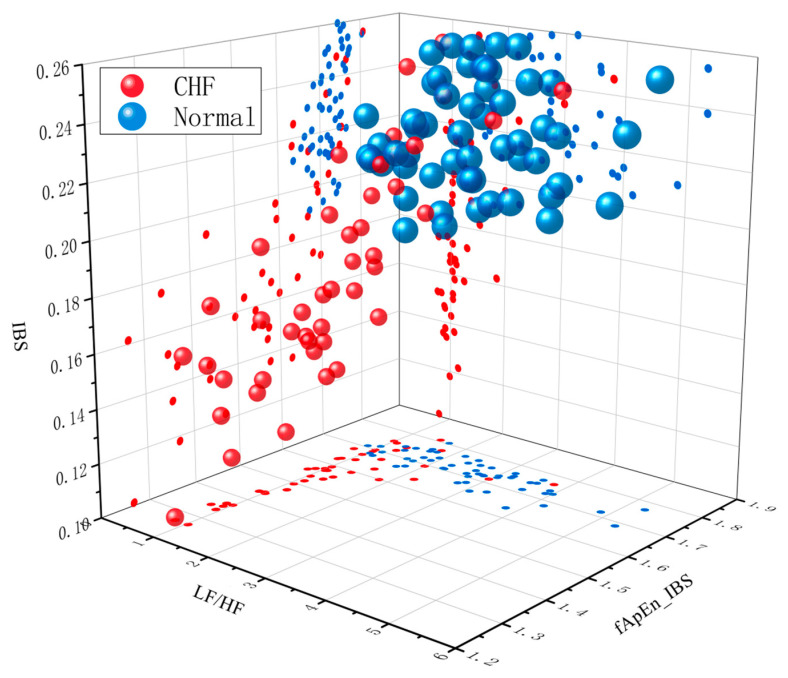
A three-dimensional image for the normal and CHF groups with LF/HF, fApEn_IBS, IBS index as the x, y, z-axis.

**Figure 7 entropy-23-01669-f007:**
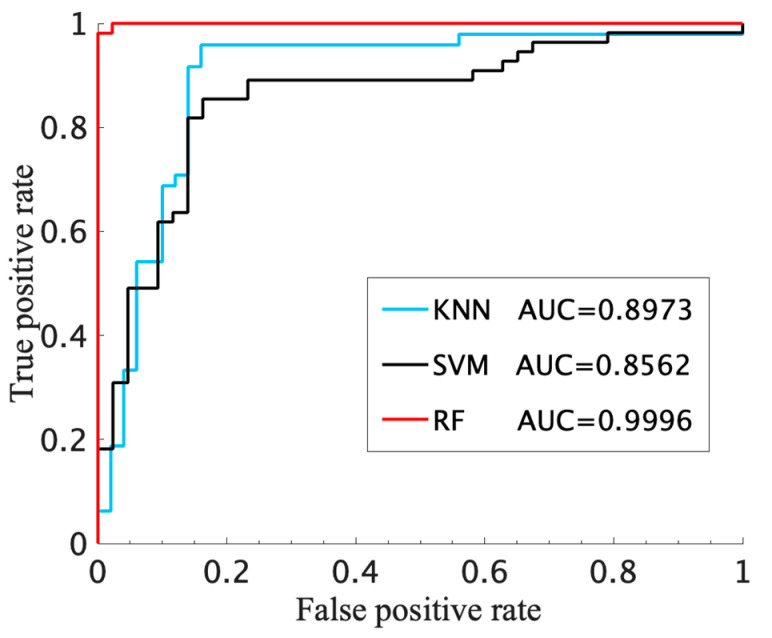
ROC curves of KNN, SVM and RF classifier for the Normal and CHF groups.

**Figure 8 entropy-23-01669-f008:**
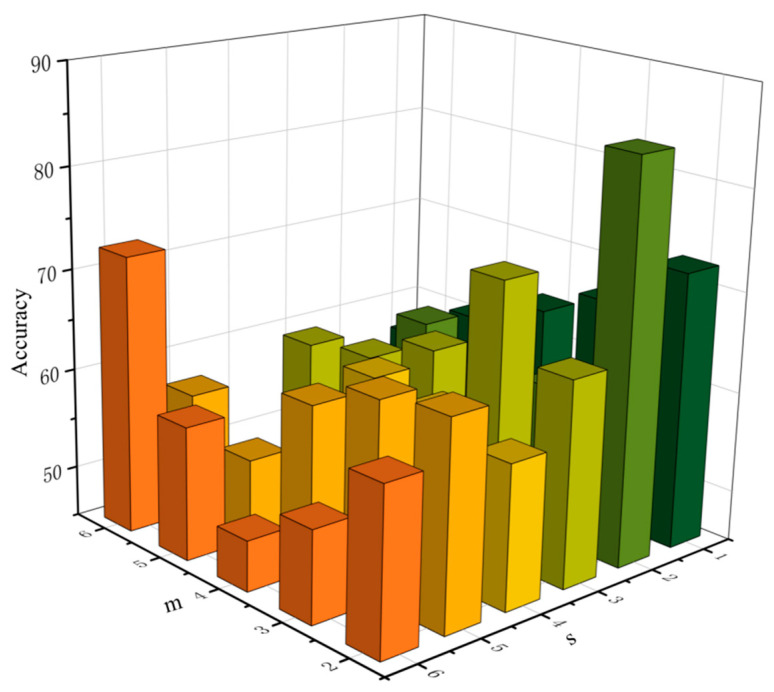
Comparison of CHF screening accuracy by the fApEn_IBS index with different m, s. fApEn_IBS: fuzzy approximate entropy of IBS index. m: the number of bits of the constructed words; s: coarse-grained scale.

**Table 1 entropy-23-01669-t001:** LF/HF, IBS and changes in the similarity of HR fluctuations indices for the CHF and normal groups.

When m = 2; s = 2			
Indices	CHF	Normal	*p*-Value
LF/HF	1.2408 ± 0.57942	2.6092 ± 1.06902	0.000 ***
IBS	0.1770 ± 0.03744	0.2273 ± 0.01874	0.000 ***
fApEn_IBS	1.5710 ± 0.14279	1.7229 ± 0.03511	0.000 ***

LF/HF: low frequency (LF)/high frequency (HF) ratio; IBS: information-based similarity index; fApEn_IBS: fuzzy approximate entropy of IBS index. CHF: CHF group; Normal: normal group. *** represents *p* < 0.001.

**Table 2 entropy-23-01669-t002:** Performance comparisons of CHF screening indicators by Fisher linear discriminant function.

Indices	TN	TP	FN	FP	Acc (%)	Sen (%)	Spe (%)
LF/HF	38	38	6	16	77.55	86.36	70.37
IBS	49	33	11	5	83.67	75	90.74
fApEn_IBS	53	30	14	1	84.69	68.18	98.15

LF/HF: low frequency (LF)/high frequency (HF) ratio; IBS: information-based similarity index; fApEn_IBS: fuzzy approximate entropy of IBS index. TP: true positive; TN: true negative; FP: false positive; FN: false negative; Acc: accuracy; Sen: sensitivity; Spe: specificity.

**Table 3 entropy-23-01669-t003:** Performance comparisons of CHF screening indicators by Fisher linear discriminant function.

Indices	Classifier	TN	TP	FN	FP	Acc (%)	Sen (%)	Spe (%)
LF/HF	RF	54	43	1	0	98.98	97.73	100
IBS	KNN	48	43	1	6	92.86	97.73	88.89
fApEn_IBS	SVM	48	36	8	6	85.71	81.82	88.89

LF/HF: low frequency (LF)/high frequency (HF) ratio; IBS: information-based similarity index; fApEn_IBS: fuzzy approximate entropy of IBS.

**Table 4 entropy-23-01669-t004:** Comparison of classification results between previous studies and our method.

Reference	Feature	Number of Recordings	Length of RR Segment	Classifier	Classification Result
Pan et al. [25]	Single-feature—MFCN (Multi-Frequency Components Entropy)	98	5 min	Fisher Discriminant	Acc = 86.7% Sen = 79.5% Spe = 92.6%
Luo et al. [51]	Multi-feature—LF/HF+TE(LF→HF)+TE(HF→LF) Transfer Entropy (TE)	98	5 min	Fisher Discriminant	Acc = 83.7% Sen = 86.4% Spe = 81.5%
David et al. [52]	Multi-feature—Renyi Entropy+SDNN+RMSSD	33	13 min	Nearest Neighbor	Acc = 87.9% Sen = 80.0% Spe = 94.4%
Chen et al. [53]	Multi-feature—50 features	116	5 min	A two-layer deep neural network model based on an SAE-based DL algorithm	Acc = 72.4% Sen = - Spe = -
Wang et al. [54]	No need	101	73.9 s	Long Short-Term Memory	Acc = 85.1% Sen = 73.6% Spe = 91.8%
Other method	Multi-feature—fApEn_IBS+IBS+LF/HF	98	1 min	Random Forces	Acc = 99.0% Sen = 97.8% Spe = 100.0%

## Data Availability

Not applicable.

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
