# Peer review of "Similarity Changes Analysis for Heart Rate Fluctuation Regularity as a New Screening Method for Congestive Heart Failure"

_entropy, 2021, doi:10.3390/e23121669_

Round 1
Reviewer 1 Report
The article is well structured and the abstract is clear enough.
The authors propose an original method for CHF screening which is based on: i) the derivation of a novel index (i.e., the fApEn_IBS index) from the combination of information-based similarity (IBS) and fuzzy approximate entropy (fApEn) methods; ii) the combination of the above index with IBS and LF/HF indexes through a machine learning classifier.
The advantage of the article is: they find the combination of parameters IBS, fApEn_IBS, and LF/HF for reaching the accuracy of 98.98%
As a general comment, the authors provide enough detailed information on the employed methods and obtained results, and the conclusive results in terms of accuracy (99%), sensitivity (97.8%) and specificity (100%) are really interesting.
However, in the reviewer's opinion some points should be better addressed by the authors in order to further improve the quality of the manuscript:
1) in the Introduction, it would be opportune to discuss the role of machine learning classifiers in the CHF detection (in this regard, the authors only provide some reference to the state of the art in Section 2, page 6, lines 189-190);
2) page 4, lines 134-141: Step 1 of the calculation procedure must be described more clearly;
3) page 7, lines 209-218: in terms of sensitivity, the results obtained by employing the fApEn_IBS index are a bit poor, and the authors should at least make some comment on this;
4) page 8, lines 223-230: Table 3 is missing, so that it is impossible to compare results obtained by different machine learning classifiers.
last consideration:
Authors should evaluate the clinical situation of CHF patients.
Sinceb Beta-blocking drugs, widely used in CHF, are known to significantly influence HRV indices,
What medications were the CHF patients taking?
Author Response
Response to Reviewer 1 Comments
Point 1: in the Introduction, it would be opportune to discuss the role of machine learning classifiers in the CHF detection (in this regard, the authors only provide some reference to the state of the art in Section 2, page 6, lines 189-190);
Response 1: I have introduced the literature to prove the feasibility of machine learning methods in the screening of patients with heart failure. Thank you for your advice.
Point 2: page 4, lines 134-141: Step 1 of the calculation procedure must be described more clearly
Response 2: I have described the steps of non-overlapping mean coarse-graining in more detail and explained again with examples.
Point 3: page 7, lines 209-218: in terms of sensitivity, the results obtained by employing the fApEn_IBS index are a bit poor, and the authors should at least make some comment on this;
Response 3: I have included my views on the results in the discussion section. (4.1Comparison and Summary). Thanks for your warning.
Point 4: page 8, lines 223-230: Table 3 is missing, so that it is impossible to compare results obtained by different machine learning classifiers.
Response 4: I have added this form, and I apologize for my carelessness. Table 3 is at the top of Figure. 7. Please check it.
Point 5: Since Beta-blocking drugs, widely used in CHF, are known to significantly influence HRV indices, what medications were the CHF patients taking?
Response 5: Although the BIDMC database noted that all of the participants were receiving only regular medication and did not take beta-blocking drugs before taking the test. However, it is undeniable that the impact of drug therapy has not been paid attention to and is not easy to quantify. Therefore, we point out this problem in the limited part and explain that future studies can analyze its impact. In addition, this paper mainly compared the HRV performance of the two groups of people. Under the same conditions, using the same non-linear model, the IBS value and fApEn_IBS value of patients were lower, reflecting the significant difference between the HRV performance of patients and normal people, and successfully revealing the physiological significance of CHF pathogenesis.

Reviewer 2 Report
The paper is well written and detailed. The topic is timely and the proposed method is intriguing and adequately explained.
Competitive baseline models are also evaluated and the proposed methods are compared to them, providing improved predictive performance.
The abstract is well defined, but it can be a bit improved to clearly show what are the contributions of the authors - proposing a new method.
Another comment is that the related work section is missing approaches for other ML methods for processing the pysionet database (see "Classification of 12-lead ECGs: the PhysioNet/Computing in Cardiology Challenge 2020", "Analysis and classification of heart diseases using heartbeat features and machine learning algorithms", "Suppression of Intensive Care Unit False Alarms Based on the Arterial Blood Pressure Signal", etc.).
Author Response
Response to Reviewer 2 Comments
Point 1: The abstract is well defined, but it can be a bit improved to clearly show what are the contributions of the authors - proposing a new method.
Response 1: We have expressed our contributions more clearly and concisely in the abstract section. Thank you for your suggestions.
Point 2: The related work section is missing approaches for other ML methods for processing the pysionet database (see "Classification of 12-lead ECGs: the PhysioNet/Computing in Cardiology Challenge 2020", "Analysis and classification of heart diseases using heartbeat features and machine learning algorithms", "Suppression of Intensive Care Unit False Alarms Based on the Arterial Blood Pressure Signal", etc.).
Response 2: In the section 3.2CHF patient screening, we added three machine learning classifiers with definitions and selection reasons to help readers better understand our intentions. And we quoted the papers you mentioned to further explain the method.
